# Integrating gene expression data into a genome-scale metabolic model to identify reprogramming during adaptive evolution

**Shaghayegh Yazdanpanah, Ehsan Motamedian** *, **Seyed Abbas Shojaosadati**

Faculty of Chemical Engineering, Department of Biotechnology, Tarbiat Modares University, Tehran, Iran

* motamedian@modares.ac.ir

## Abstract

The development of a method for identifying latent reprogramming in gene expression data resulting from adaptive laboratory evolution (ALE) in response to genetic or environmental perturbations has been a challenge. In this study, a method called Metabolic Reprogramming Identifier (MRI), based on the integration of expression data to a genome-scale metabolic model has been developed. To identify key genes playing the main role in reprogramming, a MILP problem is presented and maximization of an adaptation score as a criterion indicating a pattern of using metabolism with maximum utilization of gene expression resources is defined as an objective function. Then, genes with complete expression usage and significant expression differences between wild-type and evolved strains were selected as key genes for reprogramming. This score is also applied to evaluate the compatibility of expression patterns with maximal use of key genes. The method was implemented to investigate the reprogramming of *Escherichia coli* during adaptive evolution caused by changing carbon sources. *cyoC* and *cydB* responsible for establishing proton gradient across the inner membrane were identified to be vital in the *E. coli* reprogramming when switching from glucose to lactate. These results indicate the importance of the inner membrane in reprogramming of *E. coli* to adapt to the new environment. The method predicts no reprogramming occurs during the evolution for growth on glycerol.

## Introduction

Reprogramming of metabolism allows biological systems to adapt to new conditions caused by environmental or genetic perturbations [1]. ALE is a widely used method for investigating this reprogramming by applying artificial selection pressure in a controlled setting and long-term organism culturing over multiple generations [2]. Accumulation of beneficial mutations leads to reprogramming, especially different metabolism utilization which should be determined to understand the evolution. Due to advances in systems biology and omics technologies, the investigation of the mechanisms and dynamics of evolving metabolism has become possible using high-throughput data [3]. For instance, identifying key mutations responsible for growth improvement using whole-genome resequencing in an ALE experiment [4,5], Investigating

**Funding:** The authors received no specific funding for this work.

**Competing interests:** The authors have declared that no competing interests exist.

intracellular metabolic pathway usage by $^{13}$C-metabolic flux analysis through adaptive evolution [6], studying the reproducibility of growth phenotypes and states of global gene expression at the endpoint of adaptive evolution [7], are examples of evolution process analysis without modeling framework. In a study, underlying metabolic mechanisms that drive adaptive evolution were examined by measuring intracellular fluxes and global mRNA abundance. Except for the tricarboxylic acid cycle, no clear correlation was found between changes in gene expression and alterations in the related reaction fluxes in other pathways in central metabolism [8]. A similar result was obtained in another study that investigated metabolic mechanisms involved in the laboratory evolution of *E. coli* on lactate carbon source, which indicates the complexity of regulatory mechanisms at different levels [1]. So, the development of a powerful method is required to identify latent reprogramming in omics data and determine the genes and metabolic reactions that are the origin of the reprogramming. Constraint-based metabolic modeling approach commonly applies stoichiometric, enzyme capacity, and thermodynamic constraints to simulate biological processes. However, unlike the physicochemical sciences, biology is not only governed by physical and chemical constraints and biological functions obey genetic programs as well [9]. In fact, genetic programs similar to the software of a computer are implemented on a cell including its all chemical compounds as hardware [10]. Thus, metabolic models can be integrated with generated omics data for wild-type and evolved strains to identify the reprogramming that is hidden in these big data. Computational modeling of metabolic networks in combination with omics data can aid in determining the genetic basis of adaptation and predicting cellular response to perturbation. Several studies have been done to investigate the physiological state of evolved strains at the endpoint of adaptive evolution under diverse growth conditions, such as growth on different carbon sources [11,12] and study of the growth improvement of single-gene knockout strains [13] using flux balance analysis (FBA). pFBA is another method that examined the consistency of predicted pathway usage with expression data in evolved strains [14]. Another approach called RELATCH [15] predicted flux distribution before and after adaptation using physiological measurements and $^{13}$C metabolic flux analysis (MFA) results and gene expression data. However, all methods mentioned above required experimental uptake rates for their calculations.

In this study, we developed a method termed metabolic reprogramming identifier (MRI) for finding the key genes responsible for reprogramming to present an insightful description of the origin of evolution by defining adaptation score as criteria of compatibility of metabolism with gene expression profile pattern. The method was implemented to investigate the reprogramming of *E. coli* in an ALE experiment as a case study. Comparison of adaptation scores in wild-type and evolved strains leads to the identification of key genes that play an important role in differentiating metabolism utilization through adaptive evolution.

## Materials and methods

### Materials

In this study, the genome-scale metabolic model iJO1366 [16] was used for *E. coli*. Microarray and experimentally measured growth phenotypic data were from an adaptive evolution experiment study [7]. Seven parallel evolution experiments were conducted on two different carbon sources, L-lactate-supplemented M9 minimal medium for 60 days and glycerol-supplemented M9 minimal medium for 44 days. Gene expression in logarithmic scale and phenotypic data of wild-type strains used in this study were measured in the condition of unevolved strain transferred from glucose on either L-lactate or glycerol. For evolved strains, these data were quantified at the endpoint of evolution.

The COBRA toolbox was used to make the calculations in MATLAB software. MATLAB was linked to GAMS/CPLEX to solve LP and MILP problems [17]. The source code is freely available at http://sbme.modares.ac.ir/mri.

## The linear form of TRFBA

Transcriptional regulated flux balance analysis (TRFBA) [18] converts the metabolic model to without OR and irreversible form and extends FBA (Eq (1)) by introducing a new constraint (Eq (2)), which limits the upper bounds of reactions supported by a metabolic gene.

$$\text{Max Z} = c\upsilon$$

$$\text{S} \times \upsilon = \text{b} \tag{1}$$

$$\upsilon_{i,min} \leq \upsilon_i \leq \upsilon_{i,max}$$

$$\sum_{i \in K_j} \upsilon_i + \alpha_j = E_j \times C \tag{2}$$

In Eq (1), c is the vector of the objective function coefficients, S is an m×n stoichiometric matrix in which m is the number of metabolites, n is the number of reactions, b is the right-hand side vector determined by known reaction fluxes and $\upsilon_{i,min}$ and $\upsilon_{i,max}$ denote lower and upper bounds for the flux $\upsilon_i$. In all calculations, for intracellular reactions, an upper bound of 1000 mmol/gDCW/h is considered. Lower bounds of -1000 mmol/gDCW/h and 0 are used for reversible and irreversible intracellular reactions, respectively. In Eq (2), $K_j$ stands for the set of indices of reactions supported by metabolic gene j and $\upsilon_i$ represents reaction flux. $E_j$ is the expression level of $j^{th}$ gene and C is a constant parameter that is used to convert the expression levels to the upper bounds of the reactions in mmol/gDCW/h [18]. $\alpha_j$ indicates the amount of gene expression that is not used to produce fluxes. In fact, if all the expression of the $j^{th}$ gene is converted to flux, the $\alpha$ value corresponded to gene j would be zero which indicates the expression level of gene j is restrictive in the metabolism. Comparison of the set of genes with zero value of $\alpha$ for wild-type and evolved cells could be an indication of their difference in the pattern of using metabolism. This comparison is the basis of the method described in the next section for identifying genes that cause evolution.

## MRI representation

To determine the genes with zero value of $\alpha$, another equation was added to the model, in which genes were classified according to whether their $\alpha$ variables were zero or non-zero.

$$\alpha_j - Uz_j \leq 0, \quad \begin{cases} \alpha_j = 0 \rightarrow z_j = 0, 1 \\ \alpha_j \neq 0 \rightarrow z_j = 1 \end{cases} \tag{3}$$

The inequality shown in Eq (3) was defined for each gene with a measured expression level. The variable z is a binary variable that can only take the values of zero and one and U is a positive and big enough number. Therefore, by minimizing z values, genes that are completely used in metabolism can be obtained.

If the metabolic model is forced to maximally use the expression resources, the set of genes with zero $\alpha$ can be considered as an indication of the program that the cell had intended to run. So the comparison of these genes set for wild-type and evolved cells can guide us to find responsible genes for the reprogramming. To evaluate the cell's ability to use maximal gene

expression resources, a new parameter called adaptation score was defined, which quantifies the degree to which cell behavior matches expression data. Adaptation score indicates the number of genes whose value of the variable z is zero in a given condition which means the number of genes that are adapted to be maximally used by metabolism. To calculate the adaptation score, minimizing the sum of z was considered as the objective function according to Eq (4) and P indicates the number of metabolic genes with measured expression levels.

$$\text{Adaptation score} = \text{P} - \text{Min z} = \text{P} - \sum_{j=1}^{P} z_j \tag{4}$$

Considering minimization of sum of z values as the objective function, the model activates the forward and backward fluxes of reversible reactions and generates a cycle to unrealistically increase the number of genes with $z_j = 0$. So, to avoid simultaneous use of forward and backward directions in reversible reactions, Eq (5) to Eq (7) were added to the algorithm.

$$\upsilon_{i,f} - Uw_{i,f} \leq 0 \tag{5}$$

$$\upsilon_{i,b} - Uw_{i,b} \leq 0 \tag{6}$$

$$w_{i,f} + w_{i,b} = 1 \tag{7}$$

Here, $\upsilon_{i,f}$ and $\upsilon_{i,b}$ denote forward and backward fluxes of reversible reaction i respectively, and $w_{i,f}$ and $w_{i,b}$ are binary variables. So, if one direction of the reaction has a non-zero flux, the other one takes a value of zero.

Reactions in irreversible and without OR form [18] create similar reactions which are catalyzed by isozymes. To avoid simultaneous use of similar reactions with opposite directions and different gene associations, the above equations were written for each forward reaction with all its backwards to prevent all possible internal cycles.

To simulate the M9 minimal medium, lower bound of -1000 mmol/gDCW/h for exchange reactions including carbon source (lactate or glycerol), ammonium, water, oxygen, phosphate, sulfur, and proton were considered so that microorganism has no limit on the uptake of these compounds. The lower bound of the other exchange reactions was zero and upper bound of 1000 mmol/gDCW/h was used for all exchange reactions. Indeed, the adaptation score was calculated in the condition of unlimited rates for exchange reactions to identify the potential program that the microorganism was seeking to perform by providing a specific gene expression pattern. In this situation, the metabolic model was allowed to freely consume nutrients and secrete products and use the metabolism in a way that utilizes maximum expression resources.

If the purpose was to measure the adaptation score under the condition of maximum production of a product or maximum use of a gene, the score was calculated in the condition where the desired reaction or gene is fixed on its optimal amount. This test indicates how compatible the expression pattern is with the maximal use of a reaction or gene.

## Determination of genes involved in reprogramming

To identify genes involved in cell reprogramming through adaptive evolution from wild-type to evolved strain, as presented in Fig 1, after calculation of adaptation score under unlimited condition, two factors were evaluated. First, genes with α value of zero in wild-type strain were determined. Then, the gene expression difference of this set of genes was calculated between the wild-type strain and the mean expression of the seven evolved strains. Indeed, by considering the difference in expression, the change in the cell program in the use of that gene was

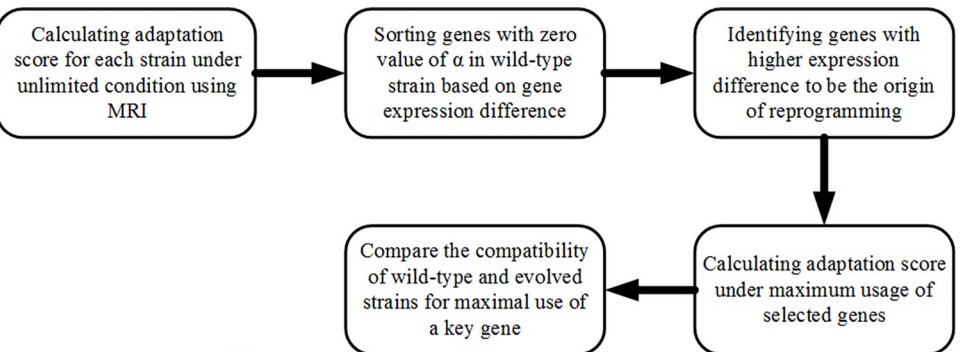

**Fig 1. Flowchart of using MRI method to identify genes with reprogrammed function in *E. coli* under adaptive evolution process.**

taken into account. Then, these genes were sorted based on the expression difference. Genes with high expression difference belonging to this gene set were considered as genes that have the potential of being the origin of the reprogramming. Thus, by calculating the adaptation score in the unlimited condition using MRI and applying these two factors, key genes that can differentiate between wild-type and evolved strains were identified. Then, the adaptation score under the condition of maximum use of each of the key genes was calculated to evaluate the compatibility of the expression pattern with the maximal use of that gene.

## Results

In this study, cell reprogramming under adaptive evolution was investigated using two adaptive evolution experimental datasets for *Escherichia coli* K-12 MG1655 strain. Datasets contained *E. coli* transferred from glucose to either L-lactate and glycerol as wild-type strain, and seven evolved populations for each carbon source (Lac2, Lac3, LacA, LacB, LacC, LacD, and LacE for lactate carbon source and Gly1, Gly2, GlyA, GlyB, GlyC, GlyD and GlyE for glycerol as carbon source) generated from daily passage of cultures into a fresh medium in the exponential growth phase [7]. Using MRI, Eq (2) (added 1353 variables to the model, Eq (3) (and set of equations related to reversible reactions (Eq (5) to Eq (7)) added 1353 and 2399 binary variables respectively to the *E. coli* model.

To determine the optimal value for parameter C, a sensitivity analysis was performed according to the method presented in ref. [18]. The sensitivity of the growth error with respect to the C values was investigated by altering the parameter and measuring the error for all evolved and wild-type strains. The optimal values of C for lactate and glycerol carbon sources were calculated which are presented in S1 Fig in S1 File.

### Investigating cell reprogramming during adaptive evolution and identification of effective genes in the lactate medium

Gene expression of a lot of genes is up and down regulated during evolution, so a method for finding the key genes having significant expression change and are effective on cell phenotype is required to identify the latent reprogramming in omics data. As presented in Fig 2, at the starting point of adaptive evolution, the wild-type strain chooses a program referred to program (1) to use its metabolism and expression resources. At the end of evolution, the expressions of genes may vary during a change from the original state due to the adaptation process, leading to program (2) for cell growth and survival. Both of these programs must be feasible and located within the solution space defined by the constraint-based model.

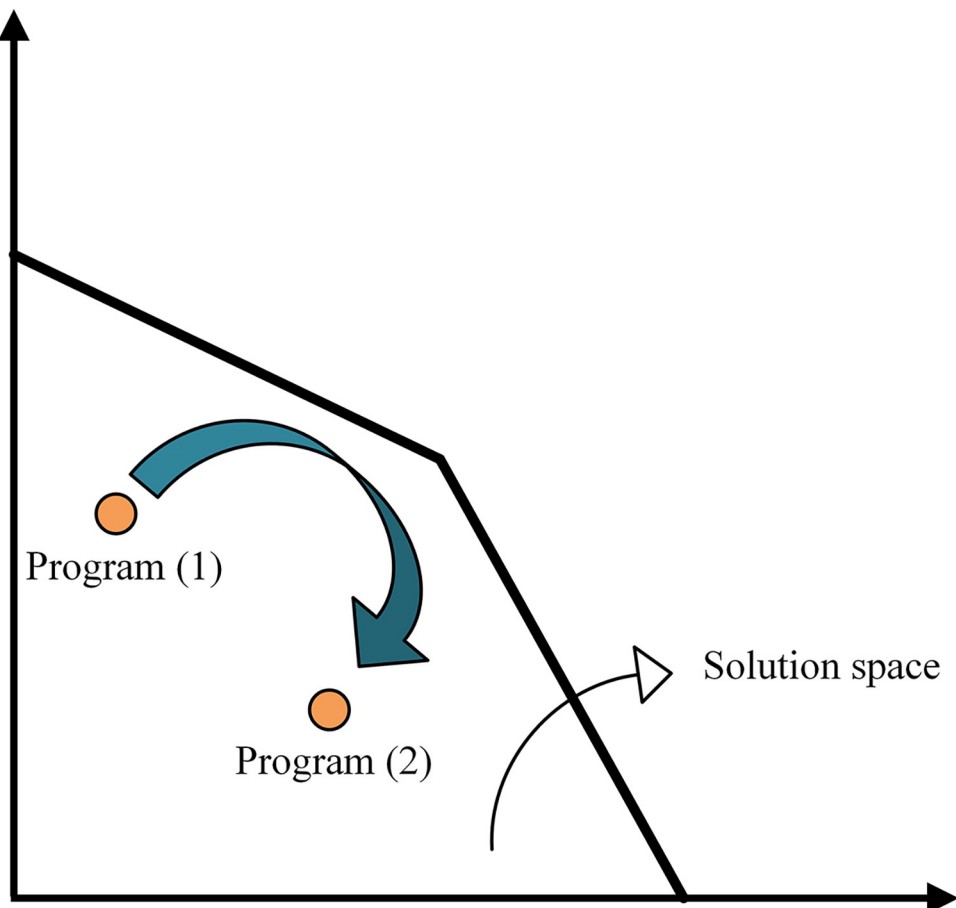

**Fig 2. Change of cell program from program (1) (wild-type) to program (2) (evolved strain) during the evolution process.**

By applying two factors of determining genes with zero value for the α variable in the wild-type strain in unlimited condition and gene expression change between wild-type and evolved strains, according to Fig 3, genes in which their usage in metabolism was changed during evolution were identified.

Of all the genes of wild-type strain that used their maximum expression in metabolism, *cyoC*, and *cydB* had the highest expression difference between the first and second states. The expression level of *cyoC* and *cydB* has increased and decreased, respectively, in the evolved strains. Indeed, not only the mean expression of *cyoC* for the seven evolved strains has increased compared to the wild-type strain but also the expression level of the wild-type strain is minimal. For *cydB*, the maximal value belongs to the wild-type strain in addition to the reduced expression levels for the evolved strains. Both genes produce a proton motive force (pmf) by transferring protons across the inner membrane. *cyoC* has higher yield with pumping 2 protons per electron while *cydB* transfers 1 proton/electron. *cyoC* is a subunit of Cytochrome bo$_3$ ubiquinol terminal oxidase which is a component of the aerobic respiratory chain of *E. coli*. The average expression of *cyoC* increased during adaptive evolution, reaching from 8.42 in wild-type to the average of 10.18 in evolved strains and the z value for this gene was zero for all strains except for the LacB sample. For *cydB* which is a subunit of Cytochrome bd-I ubiquinol oxidase and also has a role in the aerobic respiratory chain, gene expression decreased from 12.24 to 10.60 at the endpoint of evolution and only two evolved strains (Lac3 and LacC)

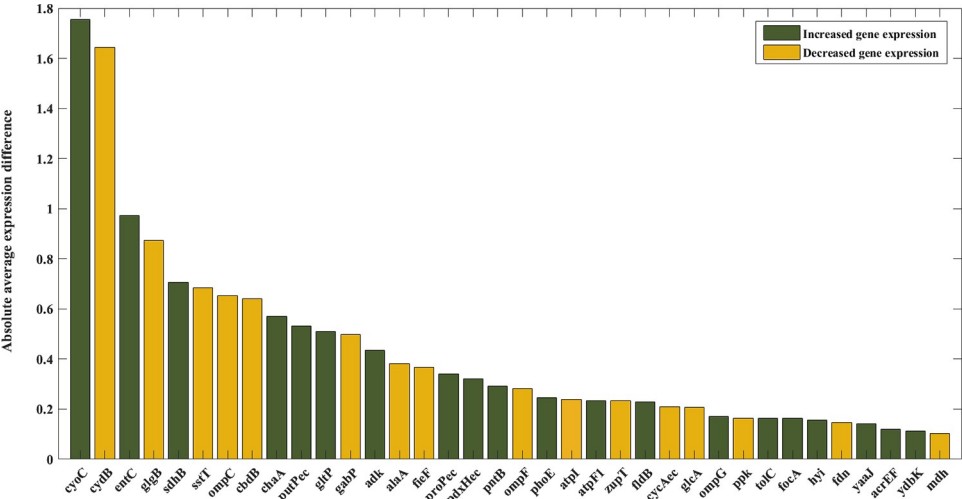

**Fig 3. Absolute average gene expression difference between wild-type and evolved strains of genes with zero value for the z variable in wild-type strain under the unlimited condition in the culture with lactate as carbon source.**

converted all the expression into flux. Decreased gene expression of *cydB* at the end of adaptive evolution in all evolved strains and incomplete utilization of this gene in 5 evolved strains under unlimited condition, indicate that *cydB* and *cyoC* function changes are related to each other. In fact, to achieve the goal of increasing pmf, in addition to increasing the expression of *cyoC* with high efficiency of proton gradient, *E. coli* decreased the expression of *cydB* which is less efficient in proton transfer through the inner membrane. Overexpression of *cyoC* and underexpression of *cydB* indicate that in the process of reprogramming, the cell is trying to increase pmf considering the higher pumping yield of *cyoC*.

The differentiation of the expression level of the wild-type strain from the evolved strains is not observed for the other genes in Fig 3. For example, while the means gene expression of the evolved strains for *entC* (the third gene in Fig 3) is about 0.97 more than the expression level of wild-type, the minimal expression level belongs to the evolved strain lacC, not wild-type. So, the two first genes in Fig 3 (*cyoC* and *cydB*) have more potential of being the origin of reprogramming. However, for more evaluation, the adaptation score under the condition of maximum use of each gene was calculated and the correlation between this score and the growth rate was assessed. The results indicated that only for *cyoC* and *cydB*, the adaption score of wild-type is differentiated from the other strains (as presented in the next sections), and for the other genes, similar to the gene expression level, the wild-type score has a value between the adaptation scores for the evolved strains and hence, they cannot be the origin of reprogramming considering that the evolved strains have significantly higher growth rate compared to wild-type.

**Assessing the role of *cyoC* in the cell reprogramming.** Considering the increase in gene expression of *cyoC* during evolution and the full use of this expression change under the condition of unlimited rate in seven strains out of eight, it can be concluded that *E. coli* has made changes in the use of this gene during the process of adaptation to improve pmf. To further investigate the reprogramming of bacteria in the use of *cyoC* during evolution, an analysis was performed in which the adaptation score of all eight strains was calculated in the condition where corresponded reaction to *cyoC* (CYTBO3_4pp) was fixed on its maximum amount to assess the compatibility of the gene expression profile to maximum use of this gene. The results of *cyoC* adaptation score calculation are given in Table 1.

**Table 1. Adaptation score of wild-type and evolved strains under the condition of maximum *cyoC* expression usage in the culture with lactate as carbon source.**

| Strain | Lac2 | Lac3 | LacA | LacB | LacC | LacD | LacE | WT |
|---|---|---|---|---|---|---|---|---|
| *cyoC* adaptation score | 61 | 62 | 51 | 60 | 60 | 53 | 56 | 37 |

*cyoC* adaptation score in each strain represents the number of genes that use their expression completely in the condition where maximum *cyoC* expression was used which is an indication of the degree of compatibility of gene expression profile with the *cyoC* expression level. According to Table 1, the *cyoC* adaptation score in the wild-type strain is much lower than the evolved strains. This indicates that the expression profile of the wild-type strain is less compatible with the maximal use of *cyoC*. However, after the evolution process, *E. coli* adaptes to the maximum use of this gene. Increased expression of this gene in evolved strains can also confirm that *E. coli* tended to use *cyoC* after adaptation. It is noticeable that although LacB uses a small part of *cyocC* expression under the condition of unlimited rate, its expression profile is highly compatible with the maximal use of *cyoC*. It should be mentioned that among evolved strains, LacE had both the lowest growth rate and yield, and LacD and LacC had the highest growth rate and yield, respectively [7]. To demonstrate changes in the pattern of bacterial metabolism usage and cell reprogramming for *cyoC* utilization through adaptive evolution, a Venn diagram was plotted in Fig 4 to compare genes having z = 0 under the condition of maximum *cyoC* usage for these three strains and wild-type.

According to Fig 4, in the wild-type strain, except for one gene, the rest of the genes are distributed in the evolved strains. However, evolved strains use all the expression of new genes after evolution and only 22 genes are in common between wild-type and all evolved strains. It is reasonable because each evolved strain has a unique evolution. For example, the LacE and LacD used 8 genes and the LacC used 5 genes, while the other strains did not use them completely. In addition, Fig 4 demonstrates that the similarity between evolved strains is more than that between wild-type and evolved strains. There are also 12 common genes between the

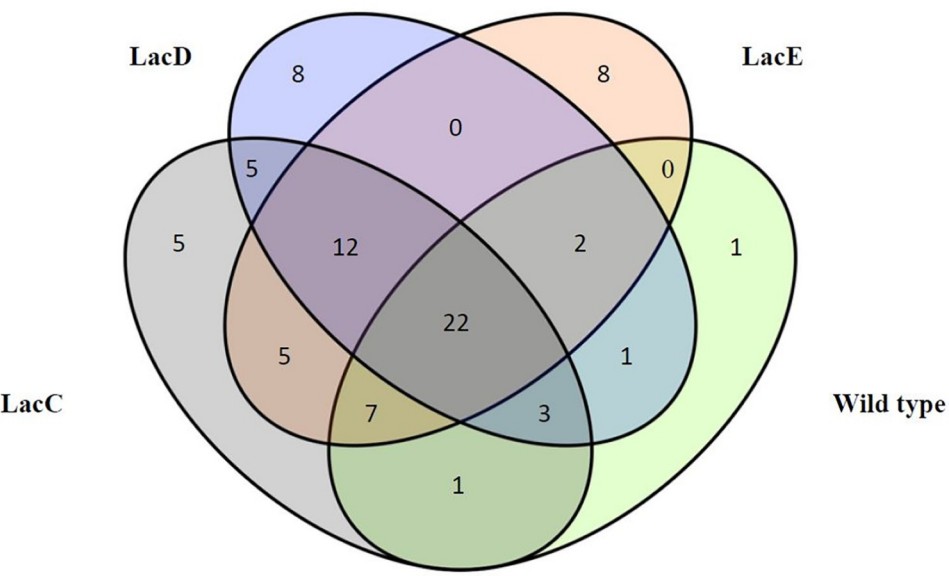

**Fig 4. Venn diagram under the condition of maximum use of *cyoC* for wild-type, LacC, LacD, and LacE in the culture medium containing lactate as carbon source.** Numbers indicate the number of genes that all of their expressions are used.

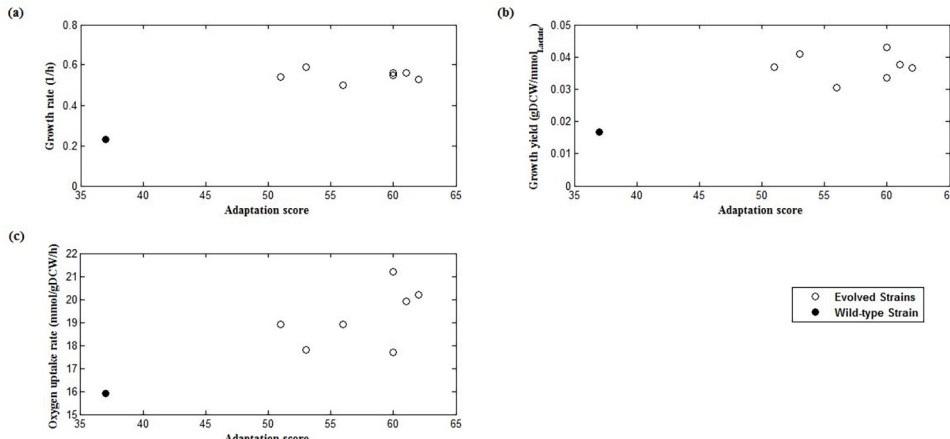

**Fig 5. Correlations between experimental phenotypic data from culture with carbon lactate source [7] and *cyoC*
adaptation score calculated using MRI.** Plots of (a) growth rate, (b) growth yield, and (c) oxygen uptake rate from
experimental data versus adaptation score under the condition of maximum *cyoC* usage (see Materials and Methods
for calculations). Open circles denote evolved strains at the endpoint of adaptive evolution and solid circles represent
wild-type strain.

three evolved strains that used all of their expression in cell metabolism. All of these could be
an indication of the difference between programs of the wild-type and evolved strains in *cyoC*
utilization.

To examine the relationship between adaptation score and growth phenotypes, the *cyoC*
adaptation score diagrams in terms of experimental growth rate, growth yield (gDCW/ mmol
lactate), and oxygen consumption were plotted (Fig 5). It can be seen that the adaptation score
can successfully show the cell reprogramming in *cyoC* usage and the wild-type strain is
completely separated from the evolved strains. In addition, Pearson correlation coefficient val-
ues (Table 2) show a direct and significant relationship between the adaptation score and each
of the experimental data (p-value < 0.05). The high and significant value of the Pearson Coeffi-
cient is because of the very different values of both the adaptation score and phenotypic data
for wild type compared to the evolved strains. Indeed, the difference between the adaptation
score of the wild-type and evolved strain and the significant relationship of the scores with
growth rate shows that this gene has been reprogrammed during the evolution.

Only gene expression data was used in this method and no other experimental uptake rates
were given as input to the model. So, this result is an important achievement, because it relates
the adaptation score which is obtained from genotypic data to growth phonotypic results.
According to Fig 5A and 5B, adaptation score is directly related to growth rate and growth
yield. Therefore, it can be concluded that maximum use of *cyoC* is associated with increased
growth rate and increased expression of this gene has an important role in bacterial growth
rate due to evolution. There is a similar pattern in the relationship between adaptation score

**Table 2. Pearson coefficient between adaptation score calculated under the condition of maximum *cyoC* use and
experimental data [7] obtained from the culture with lactate as carbon source.**

|  | Growth rate (1/h) | Growth yield (gDCW/mmol Lactate) | Oxygen uptake rate (mmol/gDCW/h) |
|---|---|---|---|
| Pearson coefficient | 0.84 | 0.76 | 0.79 |
| p-value | 0.0082 | 0.0260 | 0.0190 |

and oxygen uptake rate (Fig 5C). In evolved strains that have a growing tendency to use *cyoC*, oxygen (the final electron acceptor of the electron transport chain) consumption has also increased.

Two possibilities were suggested to explain the shift in the usage of the *cyoC*. First, the use of this gene may have evolved due to the increased need for energy production in the cell for more growth, since the main role of *cyoC* is to create a proton gradient on both sides of the inner membrane, which ultimately leads to the production of energy by ATP synthase. Another hypothesis is that since lactate transportation across the inner membrane occurs via the proton symport mechanism [19], this change may have been due to facilitating lactate entry.

In contrast to the adaptation values for the *cyoC* gene, adaptation scores under the condition of maximum flux of lactate transport reaction (lactate adaptation score) did not differ much between strains. Furthermore, the adaptation scores under the condition of maximum energy production by the ATP synthase reaction did not differentiate between the wild-type and the evolved strains (S1 Table in S1 File). Given the increased lactate uptake rate in the evolved strains based on experimental data and the compatibility of expression profiles of evolutionary strains using the *cyoC*, it can be said that the availability of protons for lactate transport through the inner membrane was a limiting factor. So, *E. coli* made changes in the use of *cyoC* to increase pmf for lactate transportation.

By dividing the lactate adaptation score by *cyoC* adaptation score to further investigate the role of increasing pmf in the carbon source consumption, it was observed that for evolved strains that had higher growth rates and yields compared to wild-type, this ratio is lower. S2 Fig in S1 File and Pearson coefficients in S2 Table in S1 File reveal that there is a strong inverse relationship between growth rate and yield with the ratio of lactate to *cyoC* adaptation score. That is, evolved strains can consume less lactate in return for the same ability of the proton pump which can result in higher efficiency for growth in comparison with wild-type. Therefore, the high adaptation under the condition of maximum use of *cyoC* to create proton gradient in evolved strains is equivalent to the optimal use of substrate and achieving a high growth rate and yield.

Using experimental substrate uptake rate instead of lactate adaptation score (S3 Fig in S1 File) also confirms that the bacteria in the evolved strains have adapted to the use of lactate as a carbon source. Pearson coefficients are presented in S3 Table in S1 File. Therefore, it can be concluded that adaptive evolution begins with a change in the carbon source of the culture medium and the reprogramming occurs through a proton gradient on both sides of the cytoplasmic membrane, adapting the bacteria to the optimal usage of the new culture including lactate instead of glucose.

**Assessing the role of *cydB* gene in the cell reprogramming.** The wild-type strain has a higher adaptation score under the condition of maximum usage of *cydB* than the evolved strains as shown in Table 3, indicating that the wild-type strain's expression pattern is more compatible with the use of this gene. The importance of *cydB* gene in cell reprogramming and adaptation score as a representative of this change is shown in Fig 6, where the wild-type strain is separated from the evolved strains. According to the Pearson correlation coefficients presented in Table 4, the inverse relationship between adaptation score and growth rate and yield

**Table 3. Adaptation score of wild-type and evolved strains under the condition of maximum *cydB* expression usage in culture with lactate as carbon source.**

| Strain | Lac2 | Lac3 | LacA | LacB | LacC | LacD | LacE | WT |
|---|---|---|---|---|---|---|---|---|
| *cydB* adaptation score | 47 | 50 | 53 | 50 | 55 | 41 | 57 | 59 |

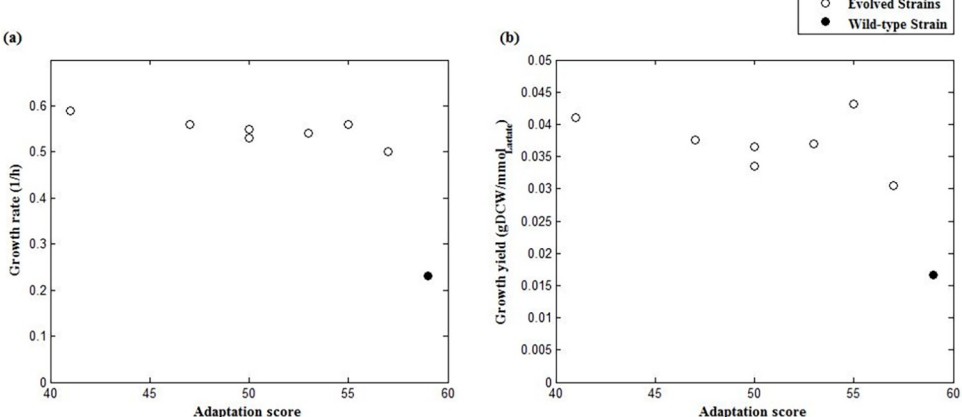

**Fig 6. Correlations between experimental phenotypic data from the culture with lactate as carbon source [7] and *cydB* adaptation score.** Plots of (a) growth rate and (b) growth yield versus adaptation score under the condition of maximum *cydB* usage. Open circles denote evolved strains at the endpoint of adaptive evolution and solid circles represent wild-type strain.

shows that evolved strains whose expression profiles do not match the maximum utilization of the *cydB* have higher growth rates and yields (p-value < 0.05). This is supported by the inverse relationship between the expression of *cyoC* and *cydB*.

Both of the genes that are shown to be important in bacterial reprogramming during the adaptive evolution process act on the inner membrane of *E. coli*. Increasing the ability of bacteria to form a proton gradient after evolution is equivalent to the optimal transport of lactate through the cytoplasmic membrane and achieving high growth rates and yields. This result could be an indication of the importance of the inner membrane in *E. coli* reprogramming. The Inner membrane plays a critical role in cellular functions as its restricted surface could limit metabolic activity [20]. In this study, the membrane occupancy constraint leads to the utilization of a more efficient reaction and decreases the activity of the gene with a lower yield in creating pmf. In most studies examining evolutionary changes, the main focus is on the central carbon metabolism and the role of membrane proteins is less considered. In another study on adaptive evolution of *E. coli* on lactate carbon source which its data is used in this study, metabolic mechanisms involved in laboratory evolution of *E. coli* on lactate carbon source were investigated based on intracellular flux states determined from $^{13}$C tracer experiments and $^{13}$C-constrained flux analysis. Most transcriptional expression and changes in metabolic flux did not exhibit clear qualitative relationships which shows complicated regulatory mechanisms [1]. The developed method in this research is capable of evaluating the effect of metabolism and membrane proteins simultaneously and finding the key genes playing the main role in the phenotype changes of evolutionary strains and reducing the complexity of understanding evolution.

**Table 4. Pearson coefficient between adaptation score calculated under the condition of maximum *cydB* use and experimental data [7] obtained from the culture with lactate as carbon source.**

|  | Growth rate (1/h) | Growth yield (gDCW/mmol$_{Lactate}$) |
|---|---|---|
| **Pearson coefficient** | -0.71 | -0.71 |
| **p-value** | 0.0465 | 0.0465 |

## Investigating cell reprogramming during adaptive evolution in the glycerol medium

For parallel cultures under adaptive evolution on a culture medium containing glycerol as a carbon source, the method did not lead to identifying genes that had been reprogrammed during evolution, as it was mentioned earlier that little changes were observed in the expression profile of the evolved strains on this carbon source [21]. This can be because of the conversion of glycerol to glyceraldehyde 3-phosphate as a metabolite in the middle part of the glycolysis pathway. Therefore, switching the carbon source from glucose to glycerol does not cause severe stress to the cell. However, the cell needs to activate gluconeogenesis instead of glycolysis for growth to shift from glucose to lactate. Furthermore, unlike lactate, glycerol can diffuse from extracellular to cytoplasm without the need for proton motive force [22]. The results of measured growth yields for both lactate and glycerol cultures presented in S4 Fig in S1 File show that the growth yield of wild-type strain in culture with glycerol as the sole carbon source is not significantly different from the evolved strains. However, this minor difference can be due to the increase in the reaction rate in the metabolism of the evolved strains, and MRI predicts that the metabolic pathway usage or pattern of using metabolism has not been changed. But, in the case of lactate, the growth yield is obviously increased at the endpoint of evolution. These observations could confirm that changing the culture medium from glucose to glycerol, probably does not make noticeable changes in the metabolism utilization during the evolution.

## Differentiating wild-type and evolved cells of other datasets using MRI

For more evaluation of this method to indicate metabolic reprogramming during adaptive evolution process, MRI was performed on four different gene expression datasets of adaptive laboratory evolution experiments of *E. coli* under elevated temperature stress and gene knockout as initial perturbation [23]. Calculation of adaptation score represented in Fig 7 showed that the adaptation score can demonstrate the differentiation between the wild-type and the evolved strains, which indicates the reprogramming occurred during the adaptation process. The ability of this method to distinguish between wild-type and evolved strains and identification of critical genes in the reprogramming of a cell under adaptive evolution can be used in further studies related to understanding the evolution as well as harnessing the reprogramming for strain design purposes.

## Discussion

Cells are capable of changing their metabolism and pathways utilization in response to genetic or environmental perturbations during the process of adaptive evolution. As these differentiations rely on natural selection and reaching the stable state, investigating changes in different levels and mimicking the cell behavior in situation of facing new conditions would be valuable for strain design purpose and developing desired phenotypes. In fact, cells change their phenotype by developing new genetic programs, and like a computer, they attempt to adapt to environmental changes by redesigning new software program controlling their hardware. This reprogramming is hidden in omics data and the development of methods for mining these big data and understanding this reprogramming is required (Fig 8). In fact, a genome-scale metabolic model which is a knowledge-based and bottom-up model was applied in this research to reveal the latent reprogramming in generated omics data as top-down data. A wild-type strain produces a pattern of gene expression based on the program it chooses. A lot of genes are up or down regulated during evolution and some of the genes play the main role in cell

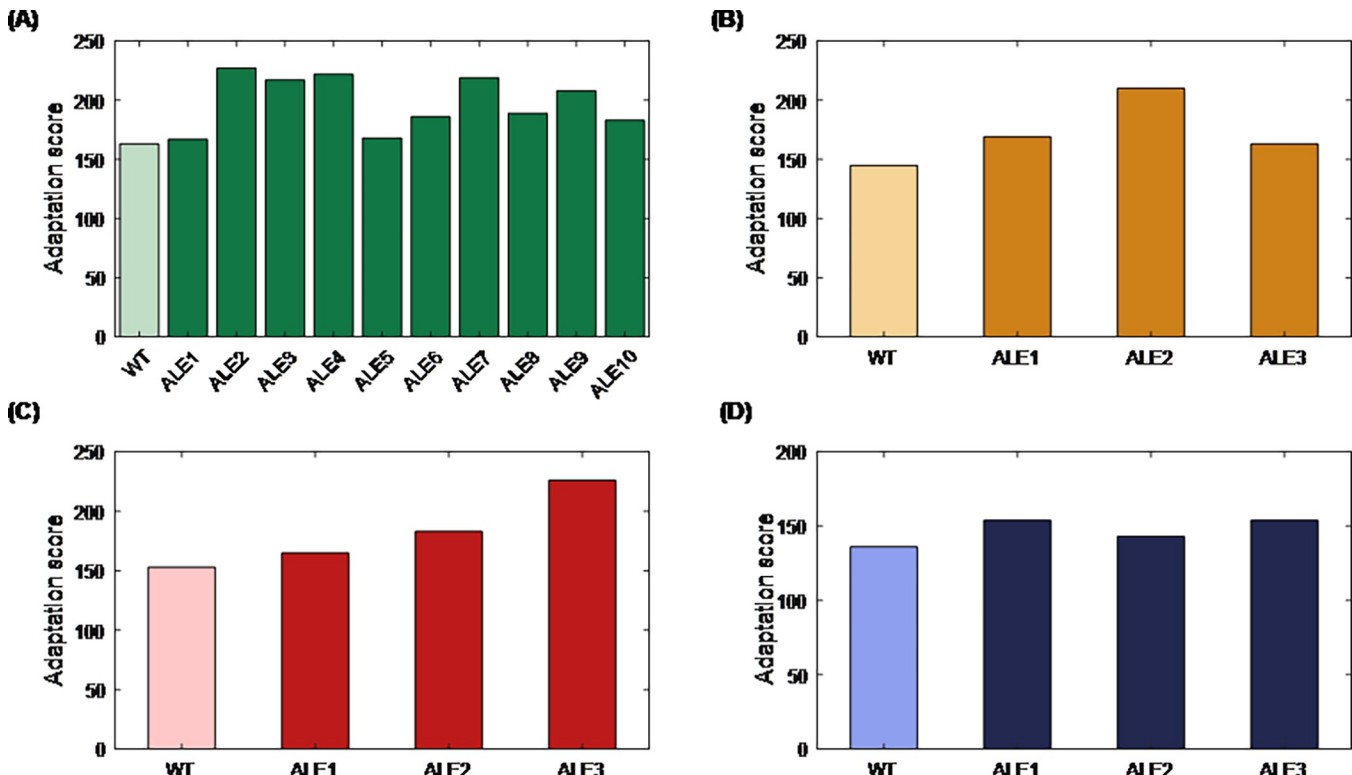

**Fig 7. Adaptation score of *E. coli* under adaptive evolution with different perturbations.** Initial perturbations are (A) temperature stress [23] (B) *ndh/cydB/appc* gene knockout [24] (C) *nuoB/cyoB* gene knockout [24] (D) *nuoB/cydB/appc* gene knockout [24].

reprogramming and phenotype change which should be identified among all transcriptomics data. Thus, in this research, a method was developed to identify the origin of metabolism changes, especially genes whose changes in expression have a meaningful effect on reprogramming. The gene expression profiles of *E. coli* at the beginning and endpoint of an ALE experiment with the change in carbon source from glucose to lactate and glycerol as environmental perturbation were integrated into the metabolic model. A MILP formula was applied to force the metabolic model to maximally use the expression resources and the adaptation score was calculated. Key genes that have the potential of being the origin of reprogramming were identified by choosing genes with higher gene expression difference between wild-type and evolved strains which have α equal to zero under unlimited condition. For evaluation of the effect of each key gene on evolution, the adaptation score under maximal use of that gene was calculated to specify whether there was a difference between compatibility of expression profiles of wild-type and evolved strains for maximal use of a gene. Then, the correlation between adaptation score and phenotypic data was calculated. For the evolution data studied in this research, two genes *cyoC* and *cydB* were identified to be reprogrammed during the adaptation process on lactate carbon source. L-lactate transportation through the inner membrane occurs via the proton symport mechanism. Given the role of these two genes in the generation of a proton motive force, it can be suggested that reprogramming in the genes' activity is related to the lactate transport reaction as the transport of lactate is coupled to proton gradient of the cytoplasmic membrane. Adaptation score under maximal use of these genes in metabolism was significantly correlated with experimentally measured growth rate and this result was one important achievement of this research because changes at genotypic level during evolution

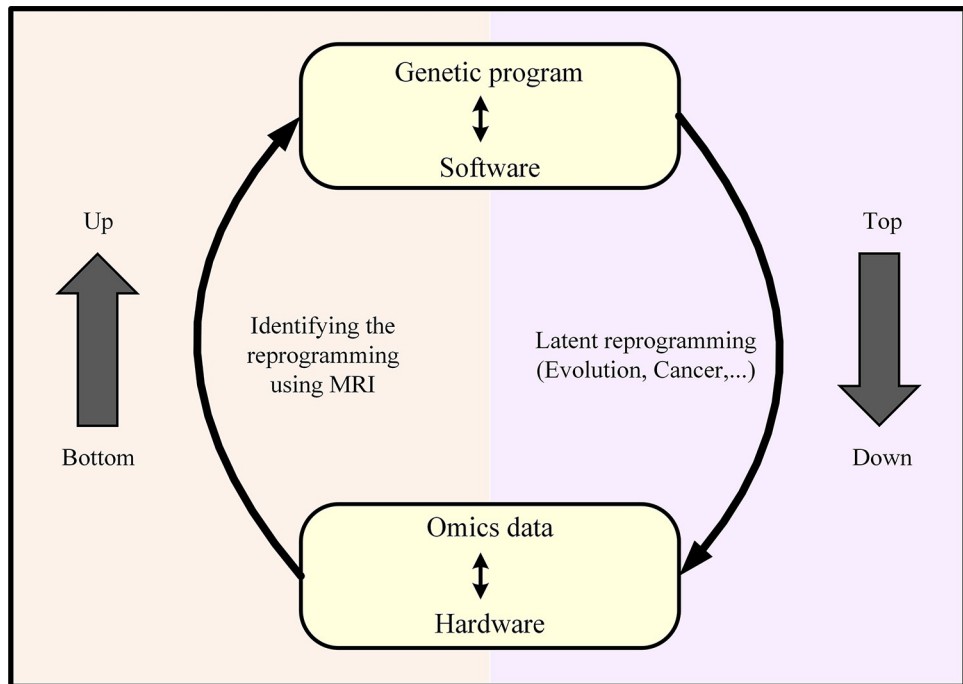

**Fig 8. A genome-scale metabolic model which is a knowledge-based and bottom-up model was applied in this research to reveal the latent reprogramming in generated omics data as top-down data using MRI.**

were correlated to changes at phenotypic level using calculated adaptation score. These results could be an evidence that microorganism increased utilization of *cyoC* to compensate for the lack of proton for lactate transportation in order to increase growth rate. However, an increase in *cyoC* usage has been associated with a decrease in *cydB* usage as a reaction with lower efficiency in proton pump. This result is consistent with the finding that a cell manages membrane proteins to achieve the highest efficiency of limited membrane's surface and a cell's activity and evolution are constrained by the physical size and structure of its proteins. As an example, *E. coli* chooses respire-fermentation which is a less efficient way for energy production, but an efficient strategy for using the limited surface of inner membrane to maximize its growth rate [25]. Moreover, increasing growth rate during the adaptive evolution process causes a reduction in the surface-to-volume (S/V) ratio [26], which is in contradiction with the cell's increasing demand for membrane space [27]. So the efficient use of the limited membrane surface would be a priority for evolved strains that are associated with increased growth rate. Generally, the developed method in this research for studying evolution indicates that the main reprogramming has occurred on cell membrane, not in intracellular metabolism.

## Supporting information

**S1 File. Supporting information.**
(DOCX)

## Acknowledgments

The authors thank Tarbiat Modares University Research and Technology Unit for supporting this study.

## Author Contributions

**Conceptualization:** Ehsan Motamedian.

**Data curation:** Shaghayegh Yazdanpanah.

**Formal analysis:** Shaghayegh Yazdanpanah, Ehsan Motamedian.

**Investigation:** Shaghayegh Yazdanpanah.

**Methodology:** Shaghayegh Yazdanpanah, Ehsan Motamedian.

**Project administration:** Ehsan Motamedian, Seyed Abbas Shojaosadati.

**Resources:** Ehsan Motamedian.

**Software:** Shaghayegh Yazdanpanah.

**Supervision:** Ehsan Motamedian, Seyed Abbas Shojaosadati.

**Validation:** Shaghayegh Yazdanpanah.

**Visualization:** Shaghayegh Yazdanpanah.

**Writing – original draft:** Shaghayegh Yazdanpanah.

**Writing – review & editing:** Ehsan Motamedian, Seyed Abbas Shojaosadati.

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
