## [Decision Letter · Decision Letter 0]

6 Aug 2023

PONE-D-23-21641Integrating gene expression data into a genome-scale metabolic model to identify reprogramming during adaptive evolutionPLOS ONE

Dear Dr. Motamedian,

Thank you for submitting your manuscript to PLOS ONE. After careful consideration, we feel that it has merit but does not fully meet PLOS ONE’s publication criteria as it currently stands. Therefore, we invite you to submit a revised version of the manuscript that addresses the points raised during the review process.

We look forward to receiving your revised manuscript.

Kind regards,

Bashir Sajo Mienda, PhD

Academic Editor

PLOS ONE

Journal Requirements:

2. We note that Figure 8 in your submission contain copyrighted images. All PLOS content is published under the Creative Commons Attribution License (CC BY 4.0), which means that the manuscript, images, and Supporting Information files will be freely available online, and any third party is permitted to access, download, copy, distribute, and use these materials in any way, even commercially, with proper attribution. For more information, see our copyright guidelines: http://journals.plos.org/plosone/s/licenses-and-copyright.

a. You may seek permission from the original copyright holder of Figure 8 to publish the content specifically under the CC BY 4.0 license. 

Reviewers' comments:

Reviewer's Responses to Questions

**Comments to the Author**

1. Is the manuscript technically sound, and do the data support the conclusions?

Reviewer #1: Yes

Reviewer #2: Yes

2. Has the statistical analysis been performed appropriately and rigorously? 

Reviewer #1: N/A

Reviewer #2: Yes

3. Have the authors made all data underlying the findings in their manuscript fully available?

Reviewer #1: Yes

Reviewer #2: Yes

4. Is the manuscript presented in an intelligible fashion and written in standard English?

Reviewer #1: Yes

Reviewer #2: No

5. Review Comments to the Author

Reviewer #1: The manuscript presents a computational method, referred to as Metabolic Reprogramming Identifier (MRI), that permits to find out which genes have undergone a significant change in their expression to adapt (for instance) to new environmental conditions. The method was tested in adaptive laboratory evolution experiments. In particular, E. coli cells were subjected to a change in the carbon source. When passing from glucose to lactate, E. coli cells appeared to reprogram their metabolism by varying the expression of two genes, cyoC and cydB, whereas no reprogramming was found by switching from glucose to galactose. Overall, I think that this is an interesting work and the MRI method might of interest for the System Biology community. I would only suggest

Reviewer #2: In this work, the authors present a method for integration of gene expression data into metabolic models, specifically to understand transcriptional adaptation following adaptive laboratory evolution. They develop an ‘adaptation score’ as a metric and maximize it using an MILP formulation. This strategy highlighted the electron transport chain as critical to the adaptation from glucose to lactate, but does not highlight any critical pathways when adapting to glycerol from glucose. The proposed mechanism of cyoC improving the proton motif force for lactate uptake when switching from glucose growth is quite interesting.

Major Comments

- In terms of presentation, the Results section came before the Methods, which resulted in not being able to follow the discussion of parameters. For example, parameter C is mentioned on line 81 but I did not even see the parameter in any of the listed equations 3-7. I would hope that the authors could add a section at the start of the Results detailing the overall structure and thinking behind the optimization formulation, with details still within the Methods. Another example is line 99: “By applying two factors of determining genes with zero value for the α variable” – this is unintelligible as the α variable has not yet been introduced. It would be preferred to at least conceptually describe what this means first.

- The method is introduced by the authors as a way to filter genes that are changing expression during adaptation down to a smaller list of the most critical genes. That said the authors highlight the electron transport chain genes as important, which from Figure 2 seem to be the genes with greatest expression changes anyways (cyoC and cydB). Other metabolic genes such as sdhB and adk are also changing expression, what would the method say about these genes? Are modeling results consistent with these changes or suggest they do not have a phenotypic effect?

- The correlations in Figures 4 and 5 in the main text and S2 and S3 in the Supporting Information seem to be influenced by the WT being very different from all evolved points. A Pearson correlation may overrate the relationship between the adaptation score and biological variable (growth rate, yield). It looks like there is very little relationship between adaptation score and rate or yield among the evolved points as well. The relationship seems stronger in Figure S3b or Figure 5a though. Could the authors comment on this? What is the argument that the score they developed is actually capturing the degree of metabolic adaptation?

- The last sentence of the abstract: the method predicts no reprogramming when switching from glucose to glycerol, but this involves the carbon catabolite repression system, which is a well-established transcription response. Could the authors comment on this?

Minor Comments

- Supporting information ‘Figure 2’ should probably be ‘Figure S2’

- Line 113: Should specify the scale of these expression values – I assume they are log2(absolute expression).

6. PLOS authors have the option to publish the peer review history of their article (what does this mean?). If published, this will include your full peer review and any attached files.

Reviewer #1: No

Reviewer #2: No

---

## [Author Response · Author response to Decision Letter 0]

11 Sep 2023

Thank you very much for your insightful comments. We revised the manuscript according to your suggestions. Here is the summary of our responses to your specified questions.

Reviewer #1:

Comments to Author(s)

The manuscript presents a computational method, referred to as Metabolic Reprogramming Identifier (MRI), that permits to find out which genes have undergone a significant change in their expression to adapt (for instance) to new environmental conditions. The method was tested in adaptive laboratory evolution experiments. In particular, E. coli cells were subjected to a change in the carbon source. When passing from glucose to lactate, E. coli cells appeared to reprogram their metabolism by varying the expression of two genes, cyoC and cydB, whereas no reprogramming was found by switching from glucose to glycerol. Overall, I think that this is an interesting work and the MRI method might of interest for the System Biology community. I would only suggest.

Reviewer #2:

Comments to Author(s)

In terms of presentation, the Results section came before the Methods, which resulted in not being able to follow the discussion of parameters. For example, parameter C is mentioned on line 81 but I did not even see the parameter in any of the listed equations 3-7. I would hope that the authors could add a section at the start of the Results detailing the overall structure and thinking behind the optimization formulation, with details still within the Methods. Another example is line 99: “By applying two factors of determining genes with zero value for the α variable” – this is unintelligible as the α variable has not yet been introduced. It would be preferred to at least conceptually describe what this means first.

To solve the problem, the two sections were replaced together. The Materials and Methods section was presented first, and the Results section came after the Materials and Methods section.

The method is introduced by the authors as a way to filter genes that are changing expression during adaptation down to a smaller list of the most critical genes. That said the authors highlight the electron transport chain genes as important, which from Figure 2 seem to be the genes with greatest expression changes anyways (cyoC and cydB). Other metabolic genes such as sdhB and adk are also changing expression, what would the method say about these genes? Are modeling results consistent with these changes or suggest they do not have a phenotypic effect?

The expression level of cyoC and cydB has increased and decreased, respectively, in the evolved strains. Indeed, not only the mean expression of cyoC for the seven evolved strains has increased compared to the wild-type strain but also the expression level of the wild-type strain is minimal. For cydB, the maximal value belongs to the wild-type strain in addition to the reduced expression levels for the evolved strains. The differentiation of the expression level of the wild-type from the evolved strains is not observed for the other genes in Figure 3. For example, while the means gene expression of the evolved strains for entC (the third gene in Figure 3) is about 0.97 more than the expression level of wild-type, the minimal expression level belongs to the evolved strain lacC, not wild-type. So, the two first genes in Figure 3 (cyoC and cydB) have more potential of being the origin of reprogramming. However, for more evaluation, the adaptation score under the condition of maximum use of each gene was calculated and the correlation between this score and the growth rate was assessed. The results indicated that only for cyoC and cydB, the adaption score of wild-type is differentiated from the other strains (as presented in the next sections), and for the other genes, similar to the gene expression level, the wild-type score has a value between the adaptation scores for the evolved strains and hence, they can not be the origin of reprogramming considering that the evolved strains have significantly higher growth rate compared to wild-type.

The explanations were added to the manuscript for more clarification.

The correlations in Figures 4 and 5 in the main text and S2 and S3 in the Supporting Information seem to be influenced by the WT being very different from all evolved points. A Pearson correlation may overrate the relationship between the adaptation score and biological variable (growth rate, yield). It looks like there is very little relationship between adaptation score and rate or yield among the evolved points as well. The relationship seems stronger in Figure S3b or Figure 5a though. Could the authors comment on this? What is the argument that the score they developed is actually capturing the degree of metabolic adaptation?

We agree with you. The high and significant value of the Pearson Coefficients is because of the very different values of both the adaptation score and phenotypic data for wild-type compared to the evolved strains. For example, if the wild-type strain is removed in Figure 4a, as shown in Figure R1, there is no clear relationship between the adaptation score and the growth rates of the evolved strains, which indicates that the cyoC has not been reprogrammed between these evolved strains. The difference between the adaptation score of the wild-type and evolved strain and the significant relationship of the scores with growth rate shows that this gene has been reprogrammed during the evolution.

More explanations were added to the manuscript for better discussion.

Fig. R1. Correlations between growth rates for the culture with carbon lactate source [1] and cyoC adaptation score for the evolved strains calculated using MRI.

The last sentence of the abstract: the method predicts no reprogramming when switching from glucose to glycerol, but this involves the carbon catabolite repression system, which is a well-established transcription response. Could the authors comment on this?

In this research, the wild-type strain is referred to as a strain that has been transferred from glucose to lactate or glycerol. In fact, the gene expression profile and phenotypic data of the wild-type strain were collected on day 0 of transfer on glycerol. Therefore, the starting point and endpoint of evolution both happens on glycerol carbon source. So we changed the sentence of the abstract to “The method predicts no reprogramming occurs during the evolution for growth on glycerol.”.

Of course, the regulation and metabolism for growth on glycerol and glucose are different. However, MRI predicted that no metabolic reprogramming has occurred during the adaptive evolution of the wild-type strain for growth on glycerol. This prediction is in accordance with the previous research [2] that observed little changes in the expression profile of the evolved strains on glycerol.

Minor Comments

- Supporting information ‘Figure 2’ should probably be ‘Figure S2’

Yes, we changed it to ‘Figure S2’ in the revised version.

- Line 113: Should specify the scale of these expression values – I assume they are log2(absolute expression).

According to the paper that presented the expression data [1], the data were analyzed with RMA using quartile normalization. The output data is in logarithmic scale. 

The scale of expression values was added to the materials and methods section.

References

1. Fong SS, Joyce AR, Palsson BØ. Parallel adaptive evolution cultures of Escherichia coli lead to convergent growth phenotypes with different gene expression states. Genome Res. 2005; 15: 1365-1372.

2. Lee D-H, Palsson BØ. Adaptive evolution of Escherichia coli K-12 MG1655 during growth on a Nonnative carbon source, L-1, 2-propanediol. Appl Environ Microbiol. 2010; 76: 4158-4168.

---

## [Editor Report · Decision Letter 1]

21 Sep 2023

Integrating gene expression data into a genome-scale metabolic model to identify reprogramming during adaptive evolution

PONE-D-23-21641R1

Dear Dr. Motamedian,

We’re pleased to inform you that your manuscript has been judged scientifically suitable for publication and will be formally accepted for publication once it meets all outstanding technical requirements.

Kind regards,

Bashir Sajo Mienda, PhD

Academic Editor

PLOS ONE
---

## [Editor Report · Acceptance letter]

25 Sep 2023

PONE-D-23-21641R1 

Integrating gene expression data into a genome-scale metabolic model to identify reprogramming during adaptive evolution 

Dear Dr. Motamedian:

I'm pleased to inform you that your manuscript has been deemed suitable for publication in PLOS ONE. Congratulations! Your manuscript is now with our production department. 

Kind regards, 

on behalf of

Dr. Bashir Sajo Mienda 

Academic Editor

PLOS ONE